

# Quantitating primer-template interactions using deconstructed PCR

Jeremy Kahsen[1,*], Sonia K. Sherwani[1,*], Ankur Naqib[2], Trisha Jeon[1], Lok Yiu Ashley Wu[1] and Stefan J. Green[1]

[1] Genomics and Microbiome Core Facility, Rush University, Chicago, IL, United States of America
[2] Rush Research Bioinformatics Core, Rush University, Chicago, IL, United States of America
[*] These authors contributed equally to this work.

## ABSTRACT

When the polymerase chain reaction (PCR) is used to amplify complex templates such as metagenomic DNA using single or degenerate primers, preferential amplification of templates (PCR bias) leads to a distorted representation of the original templates in the final amplicon pool. This bias can be influenced by mismatches between primers and templates, the locations of mismatches, and the nucleotide pairing of mismatches. Many studies have examined primer-template interactions through interrogation of the final products of PCR amplification with controlled input templates. Direct measurement of primer-template interactions, however, has not been possible, leading to uncertainty when optimizing PCR reactions and degenerate primer pools. In this study, we employed a method developed to reduce PCR bias (*i.e.*, Deconstructed PCR, or DePCR) that also provides empirical data regarding primer-template interactions during the first two cycles of PCR amplification. We systematically examined interactions between primers and templates using synthetic DNA templates and varying primer pools, amplified using standard PCR and DePCR protocols. We observed that in simple primer-template systems, perfect match primer-template interactions are favored, particularly when mismatches are close to the 3′ end of the primer. In more complex primer-template systems that better represent natural samples, mismatch amplifications can dominate, and heavily degenerate primer pools can improve representation of input templates. When employing the DePCR methodology, mismatched primer-template annealing led to amplification of source templates with significantly lower distortion relative to standard PCR. We establish here a quantitative experimental system for interrogating primer-template interactions and demonstrate the efficacy of DePCR for amplification of complex template mixtures with complex primer pools.

## INTRODUCTION

The polymerase chain reaction (PCR) is a well-established tool for amplification of regions of DNA (*Saiki et al., 1985*; *Wilson, Blitchington & Greene, 1990*). When PCRs are performed to amplify templates of unequal abundance, the final pool of PCR amplicons will often have an altered ratio of templates relative to the original sample (*Polz & Cavanaugh, 1998*; *Suzuki & Giovannoni, 1996*). Such a result is referred to as 'PCR bias' and is a well-studied

Corresponding author
Stefan J. Green,
stefan_green@rush.edu

phenomenon, particularly in the context of microbial ecology (*Pinto & Raskin, 2012*; *Polz & Cavanaugh, 1998*; *Wagner et al., 1994*). *Wagner et al. (1994)* defined two broad classes of distortion of underlying template ratios including PCR selection and PCR drift. In the first category, PCR selection, PCR conditions favor certain templates, and PCR bias generated from this selection has been attributed to a broad number of factors, including (but not limited to): annealing temperature (*Green, Venkatramanan & Naqib, 2015*; *Sipos et al., 2007*), mismatches between template and primer (*Hong et al., 2009*; *Ogino & Wilson, 2002*), location of mismatches between template and primer (*Gohl et al., 2016*), and combinatorial effects of linear copying of gDNA and exponential amplification of PCR products occurring simultaneously and at different efficiencies (*Green, Venkatramanan & Naqib, 2015*). The second category, PCR drift, is caused by stochastic effects during the early stages of PCR when primer-genomic DNA template interactions dominate (as opposed to primer-amplicon interactions) (*Polz & Cavanaugh, 1998*; *Wagner et al., 1994*). *Polz & Cavanaugh (1998)* suggested that low input gDNA could lead to higher stochastic effects.

Thus, many possible sources of PCR bias exist, and many solutions to PCR bias have been attempted. These include (but are not limited to): running fewer cycles of PCR (*Kanagawa, 2003*; *Suzuki & Giovannoni, 1996*; *Wang & Wang, 1996*), reducing degeneracies in primers whenever possible (*Polz & Cavanaugh, 1998*), and using long elongation times and/or highly processive polymerases to ensure complete copying during each cycle (*Acinas et al., 2005*). In some systems, higher annealing temperatures are recommended to reduce effects of secondary structure (*Shen et al., 2007*), while in complex template systems such as microbial DNA, lower annealing temperatures are recommended to improve tolerance for mismatch annealing (*Hongoh et al., 2003*).

We previously developed the "Deconstructed PCR" (DePCR) method (*Green, Venkatramanan & Naqib, 2015*; *Naqib, Poggi & Green, 2019*) to reduce PCR bias (Fig. 1). DePCR operates by separating linear copying of templates (*i.e.*, primers annealing to the source DNA template) from exponential amplification of template copies which begins in the third cycle of amplification (*Green, Venkatramanan & Naqib, 2015*; *Naqib, Poggi & Green, 2019*). In addition to reducing bias, the DePCR method preserves information lost in standard PCR: namely, the identity of each primer annealing to source DNA templates (*Green, Venkatramanan & Naqib, 2015*; *Naqib, Poggi & Green, 2019*). In standard PCR, after cycling, the primer sequences present in the final amplification products represent primers that have annealed to amplification products over the course of 20–30 cycles of amplification, not the original template DNA. Under standard PCR, multiple different primers within a degenerate pool of primers can interact with copies during cycling. This process degrades information about which primers anneal to source DNA templates and represents a 'scrambling' of the primer-template interaction information. In this study, we ues the ability of DePCR to preserve the primer-template interactions, in tandem with known synthetic DNA templates, to explore effects of these interactions with the goal of understanding and reducing PCR bias. Although an extensive number of prior studies have examined primer-template interactions (*Lefever et al., 2013*; *Pan et al., 2014*; *Piñol et al., 2015*; *Stadhouders et al., 2010*), none of those studies was able to empirically measure
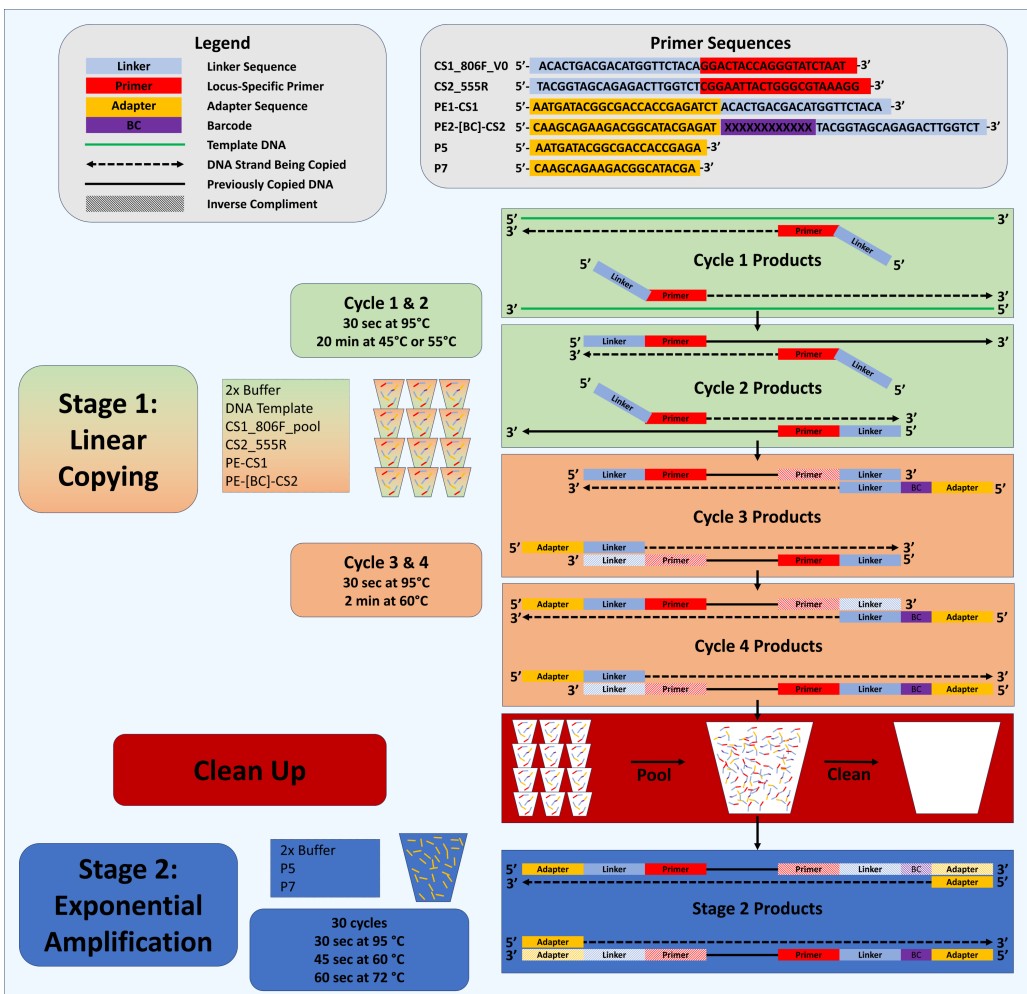

**Figure 1** **Schematic of Deconstructed PCR (DePCR) workflow.** CS1, common sequence 1 linker sequence. CS2, common sequence 2 linker sequence. BC, barcode. F, Forward primer. R, Reverse primer, P5/P7, Illumina primers, PE1/PE2, Fluidigm Access Array Barcode Library Illumina adapters. In stage 1 (linear copying only), individual samples are cycled for four cycles with locus-specific primers and Fluidigm barcoded primers. Subsequently, all reactions are pooled and purified together, and then amplified with Illumina P5 and P7 primers in stage 2 (exponential amplification with primers targeting linker sequences). During stage 1, linear copying of templates leads to products which contain Illumina sequencing adapters, sample-specific barcodes, and the locus-specific region of interest. Only fragments with Illumina adapters and barcodes are exponentially amplified in stage 2. Locus-specific primer sequences can be modified as needed.

which primer anneals to which template, and only end-point PCR product composition through sequencing or quantitative PCR was used to optimize PCR conditions. We hypothesized that DePCR could be used with synthetic DNA templates and synthesized primers to quantify primer-template interactions in systems containing mismatches. We further hypothesized that DePCR, by limiting the number of cycles of primer-template interaction, could improve representation of the source DNA templates in the final PCR product when using either degenerate or non-degenerate primers.

To explore primer-template interactions, we synthesized 10 double-stranded DNA templates with unique priming sites and 64 'forward' primers, 20 bases in length, with 0, 1, 2 or 3 mismatches with each of the 10 templates. For primers and templates with mismatches, mismatches were located close to the 3′ end of the primer (-2 position, counting from the 3′ end), the middle of the primer (-8), or closer to the 5′ end of the primer (-14). Both standard PCR amplification protocols and DePCR amplification protocols were used to amplify templates in a series of experiments in which templates, primers, and annealing temperature were varied. Amplicon sequencing was performed on an Illumina MiniSeq sequencer, generating thousands of sequences per sample for robust quantification of amplicons. Additional experiments with a proof-reading polymerase were conducted to address concerns that observed signals were the result of polymerase error.

## MATERIALS AND METHODS

Portions of this manuscript were previously published as parts of a preprint and thesis (*Naqib, 2019*; *Naqib, Jeon & Green, 2019*).

### Nucleic acids

Ten artificial double-stranded DNA (gBlocks Gene Fragments, called "synthetic templates" or ST) were synthesized by Integrated DNA Technologies, Inc. (IDT; Coralville, Iowa). Sequences of the gBlocks are provided in Supplemental Materials S1. The synthetic DNA sequences were based on a 451 bp segment of the 16S rRNA gene sequence from a Gammaproteobacterium, *Rhodanobacter denitrificans* (*Prakash et al., 2012*). The sequence was modified from the original by reducing the amplicon size so that the amplification product could be sequenced on an Illumina MiniSeq sequencer with a 300-cycle chemistry kit. Each ST gBlock was nearly identical, with variations at three positions in the forward (806F) priming site and with unique 'recognition sequences' in the middle of the gBlock for each priming site variant (Fig. 2). The recognition sequences in each gBlock were created by scrambling bases at positions 99–109 (from the beginning of the priming site), creating a minimum Hamming distance of four between any templates. For each gBlock, bases at three positions (−2, −8 and −14 bases from the 3′ end) were varied (*e.g.*, the sequence of the ST0 template 806F primer site is GGACTA**C**CAGGG**T**ATCTA**A**T, with variant positions underlined and bolded). The variant positions are labeled 5′, M (for middle), and 3′. All gBlocks were otherwise identical and had no sequence differences at the reverse (555R) primer site. Prior to pooling, each ST was quantitated using fluorimetry with a Qubit 4.0 Fluorometer with the dsDNA BR Assay (Thermo Fisher Scientific, San Jose, CA). DNA concentrations were equalized among all STs prior to pooling.

For initial experiments, 64 different standard oligonucleotide primers were synthesized as LabReady primers, normalized to 100 μM concentration (IDT). Sequences are shown in Table S1. For experiments conducted with a proof-reading polymerase, primers were synthesized as above, but with phosphorothioate bonds for the two nucleotide bases at 3′ ends to reduce nucleolytic degradation of oligonucleotides. Most synthesized primers contained so-called 'common sequences' (*i.e.,* linkers) at the 5′ ends. The linker sequences differed for forward primers (CS1 linker; ACACTGACGACATGGTTCTACA) and reverse

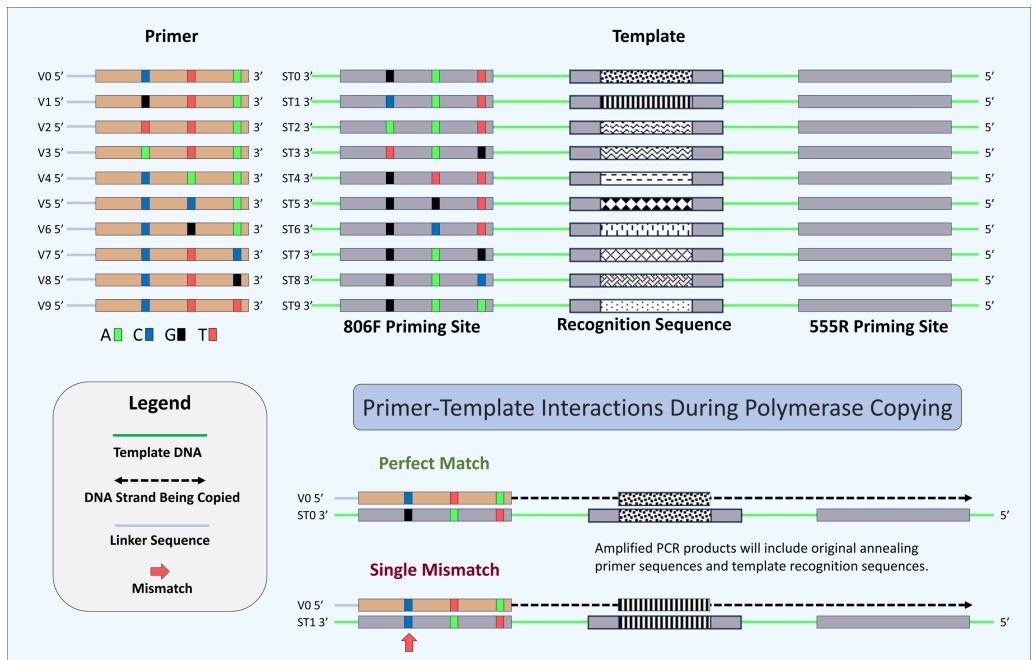

**Figure 2** **Schematic of primers, template, and mechanism for determining perfect match and mismatch annealing.** Sixty-four unique oligonucleotide primers were synthesized in this study of which ten are shown here (V0–V9). Primers were identical except for three positions at -2, -8 and -14 positions relative to the 3′ ends. Variant bases have been indicated by color (''C'', Blue, ''T'', Red, ''A'', Green, and ''G'', Black). A schematic of the ten synthetic DNA templates used in this study (ST0 to ST9) are also shown. Each template was identical except for the 806F priming site and the 12-base recognition sequence. Each unique priming site sequence is linked with a unique recognition sequence. A total of 640 potential primer-template interactions can occur in this system (10 templates × 64 primers), of which two are shown here. Shown are primer-template interactions indicating the annealing of a perfectly matched primer (template ST0 and primer V0) and a primer with a single mismatch (template ST1 and primer V0). Perfect match and mismatch annealing are determined by comparing the inferred primer site based on the recognition sequence to the observed primer sequence for each sequencing reaction. Only reactions conducted using the DePCR methodology retain the sequence of the primer annealing to the source DNA templates.

primers (CS2 linker; TACGGTAGCAGAGACTTGGTCT). For barcoding reactions, Access Array Barcode Library for Illumina primers were purchased (Fluidigm, South San Francisco, CA, USA). These Access Array primers contain Illumina sequencing adapters, a single unique 10-base index, and CS1 or CS2 linkers at the 3′ ends of the oligonucleotides. The final working concentration of Access Array primers was 400 nM. Illumina P5 (AATGATACGGCGACCACCGA) and P7 (CAAGCAGAAGACGGCATACGA) primers were synthesized without linkers or phosphorothioate modification.

The synthetic templates were combined into several different pools (*i.e.*, A, B, C, D and E) each with a different number of templates and template concentrations (Table 1). For example, in ''A'' experiments, only a single template (ST0) was employed, while in ''B'' experiments, all 10 templates were mixed equimolarly. In addition to templates, primer pools varied from a single primer matching the ST0 template (*i.e.,* primer V0) to a complex

**Table 1  Experimental pools of primers and templates.** Templates and primers were grouped into experimental pools. Synthetic templates pools were designated by letters A–E. Primer pools were designated as 1, 9, 10, 27 and 64, depending on the number of primers included in equimolar concentrations. Full details of primer-template combinations are provided in Table S2.

**Experimental pools**

Template

| Pool name | "ST"s | Relative concentrations | Description |
|---|---|---|---|
| A | ST0 | Equimolar | Original ST |
| B | ST0-ST9 | Equimolar | ST0 + 9 with a single mismatch |
| C | ST0-ST9 | ST0x0.1 + ST1-9x1 | ST0 at 10% relative concentration |
| D | ST0 + ST7-ST9 | ST0x1 + ST7x2 + ST9x4 + ST8x8 | 2-fold increases of 3′ mismatch |
| E | ST0 + ST4-ST6 | ST0x1 + ST5x2 + ST4x4 + ST6x8 | 2-fold increases of M mismatch |

Primer

| Pool name | "V"s | Relative concentrations | Description |
|---|---|---|---|
| 1 | V0 | Equimolar | Perfect match to V0 |
| 9 | V1–V9 | Equimolar | All single mismatches to V0 |
| 10 | V0–V9 | Equimolar | One perfect match for each template ST0-ST9 |
| 27 | V10–V36 | Equimolar | All double mismatches to ST0 |
| 64 | V0–V63 | Equimolar | All possible primer variants |

pool of all 64 synthesized primers (Table 1). Within each pool, primers were equimolar. In experiment 'B10', the template pool consisted of all 10 ST templates in equimolar concentrations and 10 primers, each a perfect match to one of the ST templates. Across all experiments, each primer had 0–3 mismatches with each of the 10 templates (Table S2). In Experiment A+, a series of nine pairs of forward primer mixtures were made to interrogate all possible mismatches at the three variant positions. Primer combinations included: V0 only (control); V0 + V1; V0 + V2; V0 + V3; V0 + V4; V0 + V5; V0 + V6; V0 + V7; V0 + V8; and V0 + V9.

## Standard PCR protocol

Initial experiments in this study were performed using a non-proof-reading polymerase (MyTaq HS 2X master mix; Bioline, Taunton, MA, USA), referred to in figures at "MT". Subsequent experiments were conducted with either a non-proof-reading polymerase (DreamTaq Green PCR Master Mix 2X, Thermo Scientific, Waltham, MA, USA), referred to as "DT", or with a proof-reading polymerase (Hifi ToughMix, QuantaBio, Beverly, MA, USA), referred to as "QB". PCR amplifications were conducted with either a standard, two-stage PCR amplification method ('PCR') or using the Deconstructed PCR amplification method ('DePCR').

For the initial experiments, the standard PCR method used is a two-stage PCR amplification method that produces barcoded amplicons for next-generation sequencing (*Naqib et al., 2018*). First stage PCR amplifications were performed in 10 μL reactions in 96-well plates, using HS 2X MyTaq master mix (Bioline, Taunton, MA, USA). 2.5 ng of ST mixtures (template pools A, B, C, D, and E; Table 1) were used for each 10 μL reaction. Primer pools (Table 1) were added at a final concentration of 200 nM. All reactions were performed with eight technical replicates. Thermocycling conditions were 95 °C for 5 min,

28 cycles of 95 °C for 30 s, annealing temperatures of 45 °C or 55 °C for 45 s, and 72 °C for 30 s, and a final elongation at 72 °C for 7 min. Subsequently, a second PCR amplification was performed in 10 µL reactions in 96-well plates. A master mix for the entire plate was made using the MT master mix, and each well received a separate primer pair with a unique Access Array Barcode Library for Illumina primer (described above). One µL of the first stage PCR reaction, without purification, was added to the second stage reaction. Cycling conditions were as follows: 95 °C for 5 min, followed by eight cycles of 95 °C for 30″, 60 °C for 30″, and 72 °C for 30″. A final seven-minute elongation step was performed at 72 °C. The second stage PCR amplicons were pooled together, and the pooled library was purified using an AMPure XP cleanup protocol (0.7X, vol/vol; Agencourt, Beckmann-Coulter) to remove short fragments. Pooled and cleaned amplicons were sequenced on an Illumina MiniSeq mid-output flow cell with 2 × 153 base reads, and with an approximate 30% phiX spike-in.

For our secondary experiments, including the repeat of the A1 experiment, we followed the same protocol as described above for the two-stage PCR amplification method, but with DT and QB polymerases. QB thermocycling conditions for the first stage of standard PCRs were 98 °C for 30 s, 28 cycles of 98 °C for 10 s, annealing temperatures of 45 °C or 55 °C for 5 s, and 68 °C for 1 s. QB thermocycling conditions for the second stage of standard PCRs were 98 °C for 2 min, 8 cycles of 98 °C for 10 s, annealing temperature of 60 °C for 1 s, and 68 °C for 1 s. DT thermocycling conditions for the first stage of standard PCR were 95 °C for 1 min, 28 cycles of 95 °C for 30 s, annealing temperatures of 45 °C or 55 °C for 30 s, and 72 °C for 1 min, and final elongation at 72 °C for 5 min. DT thermocycling conditions for the second stage of standard PCR were 95 °C for 2 min, 8 cycles of 95 °C for 30 s, annealing temperature of 60 °C for 45 s, and 72 °C for 90 s.

## Deconstructed PCR (DePCR) protocol

A two-stage deconstructed PCR (DePCR) method (*Green, Venkatramanan & Naqib, 2015*; *Naqib, Poggi & Green, 2019*) was also used to generate amplicons for next-generation sequencing (Fig. 1). In this protocol, four primers were added to the first stage reaction, including locus-specific primer pools containing 5′ CS1 and CS2 linkers (200 nM final concentration; Table S1). Each well also received an Access Array primer pair with a unique barcode (400 nM final concentration). 2.5 ng of synthetic ST mixtures (gBlock pools A, B, C, D and E; Table 1) was used for each 10 µL reaction. In initial experiments, reactions were performed using MT mastermix and reactions were conducted in 96-well plates. First stage DePCR thermocycling conditions were: initial denaturation at 95 °C for 5 min, followed by two cycles of 95 °C for 30 s and either 45 °C and 55 °C for 20 min, followed by two cycles of 95 °C for 30 s and 60 °C for 2 min (Fig. 1). Subsequently, technical replicates from each experiment (*e.g.*, A1, A10, B10, *etc.*) were pooled together from both annealing temperatures. Pooled replicates were purified twice sequentially using an AMPure XP cleanup protocol (0.7X, vol/vol) and eluted in 50 µL to remove primers. Of this eluate, 20 µL were used as template for amplification in the second stage reaction with Illumina P5 and P7 primers. Final volume for each amplification reaction was 50 µL. Second stage DePCR thermocycling conditions were 95 °C for 5 min and 30 cycles of 95 °C for 30 s,

60 °C for 45 s, and 72 °C for 90 s. Amplicons generated from second stage reactions were again purified using an AMPure XP cleanup protocol (0.7X, vol/vol). Pooled and purified amplicons from each experiment were quantified using Qubit fluorimetry (Qubit 4.0, Thermo Fisher Scientific), and further pooled together to generate a final library. Pooled, cleaned amplicons were sequenced on an Illumina MiniSeq mid-output flow cell with 2 × 153 or 2 × 154 base reads, and with an approximate 30% phiX spike-in. Library preparation and sequencing were performed either at the Genome Research Core (GRC; University of Illinois at Chicago) or at the Genomics and Microbiome Core Facility (GMCF; Rush University).

For additional experiments (*i.e.,* repeat of A1 with a proof-reading polymerase and A+), the DePCR method was performed as described above but replacing the MT mastermix with QB and DT mastermixes. Cycling conditions were also altered according to the requirements for each polymerase. For QB cycling, initial denaturation was performed at 98 °C for 5 min, followed by two cycles of 98 °C for 30 s and either 45 °C and 55 °C for 20 min, followed by two cycles of 98 °C for 30 s and 60 °C for 2 min. Final cycling with Illumina P5 and P7 primers was performed with QB mastermix with initial denaturation at 98 °C for 2 min, 15 cycles of 98 °C for 10 s, annealing temperature of 60 °C for 1 s, and 68 °C for 1 s. For DT cycling, initial denaturation was performed at 95 °C for 5 min, followed by two cycles of 95 °C for 30 s and either 45 °C and 55 °C for 20 min, followed by two cycles of 95 °C for 30 s and 60 °C for 2 min. Final cycling with Illumina P5 and P7 primers was performed with DT mastermix with initial denaturation at 95 °C for 5 min, 15 cycles of 95 °C for 30 s, annealing temperature of 60 °C for 45 s, and 72 °C for 90 s. Sequencing was performed on an Illumina MiniSeq as described above.

## Sequence data analysis

Raw FASTQ files were merged using the software package PEAR (*Zhang et al., 2014*) using default parameters. Merged reads were then converted from FASTQ to FASTA format using the function convert_fastaqual_fastq.py within the software package QIIME (*Caporaso et al., 2010*). Sequence data was analyzed to identify recognition sequences (*i.e.,* identifying which of the 10 templates was amplified), and to identify the sequence of the primer used to amplify the template (*i.e.,* identifying which of 64 possible 'forward' primers was used for amplification). In total, 640 possible primer-template pairs were considered, though each experiment had fewer possible combinations. A list of template sequences is provided in Supplemental Materials S1, and a list of all primer sequences is shown in Table S1. All possible primer-template interactions are shown in Table S2. To calculate utilization profiles for all the samples, a mapping file containing all possible unique combinations of 806F primers and recognition sequences was generated (Table S3). To identify the 640 unique primer-recognition sequence combinations that could occur, a custom bash UNIX shell script (Supplemental Materials S2) was written to search for each combination. Only sequences that matched perfectly with a primer variant sequence and a recognition sequence were counted. In the end, all counts were collated to generate a biological observation matrix (BIOM) (*McDonald et al., 2012*) (Table S4). The BIOM was rarefied to a depth of 7,000 counts per replicate in the R programming environment (*R Core Team,*

*2013*) for all downstream analyses. The BIOMs were further split into template BIOMs (10 features) and primer BIOMs (64 features). Heatmaps for both template and primer BIOMs were generated using the package pheatmap in R. The vegan R package (*Oksanen et al., 2011*) was used to generate alpha diversity indices and to calculate pairwise Bray–Curtis dissimilarity scores. Metric multi-dimensional scaling (mMDS) plots were created using the cmdscale and ggplot2 R package (*Wickham, 2010*). Ellipses, representing 95% confidence intervals around group centroids, were created assuming a multivariate t-distribution. Analysis of similarity (ANOSIM) calculations were performed in the software package Primer7 (Primer-E, Plymouth, UK) (*Clarke & Gorley, 2015*).

For analysis of A+ and repeat A1 experiments, a separate analysis pipeline was employed. Before analysis, raw FASTQ files were merged using the software package PEAR (*Zhang et al., 2014*) using default parameters. Merged reads were filtered to a minimum quality of 20 using the Insect R package (*Wilkinson et al., 2018*). No rarefication was performed. Sequence data were analyzed by position against the synthetic template sequences to determine error rate. Forward primers were identified by the variable regions at positions -14, -8, -2 from the 3′ end and the templates were identified by recognition sequences at positions 99–109 (Table S3). Matches not belonging to templates or primers included in each experiment were removed from the data sets. The R packages ggplot2, gridExtra and ggh4x were used to create figures (*Auguie, Antonov & Auguie, 2017*; *Van den Brand, 2024*; *Wickham, 2010*). All coding resources are in Supplemental Materials S3.

## Calculation of primer and template metrics
### Ideal Scores

To evaluate how well sequence data represented the expected outcomes based on input templates, we performed an 'Ideal Score' (IS) analysis within the vegan R package, generating 'Ideal Primer Score' (IPS) values and 'Ideal Template Score' (ITS). The IS analysis is a summation of the absolute difference between the expected relative abundance and the observed relative abundance for each feature in a multi-feature dataset and is conceptually derived from Bray–Curtis dissimilarity. The IS has a range from 0 (perfect representation of the input template distribution) to 200. The IS analysis was slightly modified from the formula described previously (*Green, Venkatramanan & Naqib, 2015*) to account for uneven distribution of templates. For templates, the ITS is a measure of dissimilarity of observed distribution of templates relative to the input ratio of templates. Similarly, the IPS represents a measure of dissimilarity of utilization of primers in amplification reactions relative to the input ratio of primers into a reaction. Higher values for ITS represent higher distortion of the underlying input DNA template distribution. Higher values for IPS represent more selective use of primers in reactions.

### Mismatch ratio

We calculated a univariate parameter to evaluate the relationship between perfect match annealing and mismatch annealing. The 'mismatch ratio' is the ratio of sequences with mismatches between its identified primer and template relative to sequences with perfectly matching primer and template. For our system any primer-templates pairs with matching numbers are a perfect match (*e.g.*, V0 and ST0), and any primer-template pairs with

non-matching numbers are mismatched (*e.g.*, V0 and ST1; V0 and ST2, *etc.*). All possible matching pairs ($N = 640$; 64 primers $\times$ 10 templates) are shown in Table S2 columns AE-BL. A value of 0 for the mismatch ratio indicates that only perfectly matching primers were utilized, and a value of 1 indicates that the utilization of perfectly matching primers and single mismatch primers was equal. The mismatch ratio scale is from 0 to infinity (*i.e.,* no perfect match interactions).

## Statistical analysis

Welch's two sample *t*-test was used to calculate *p*-values for all tables and for all error rate comparisons. No multiple comparison corrections were applied. For the analysis shown in Fig. 3, analysis of variance (ANOVA) was performed for each temperature, followed by Tukey's post-hoc test.

## RESULTS AND DISCUSSION

### Experimental design

The purpose of this study was to employ DePCR to better understand primer-template interactions during PCR amplification in controlled experiments with varying primer pools and templates. DePCR serves two roles in this study. First, it is a method that has been previously shown to reduce PCR bias (*Green, Venkatramanan & Naqib, 2015*; *Naqib, Poggi & Green, 2019*) by separating linear copying and exponential amplification, and by using non-degenerate primers that target linker sequences to perform exponential amplification instead of locus-specific degenerate primers. Secondly, DePCR also preserves a signal of which primer anneals to DNA templates during the first two cycles of PCR (*i.e.,* during linear amplification only). This is invaluable for understanding primer-template interactions in PCR amplifications employing degenerate primers. In standard PCR amplification, this signal is lost due to primers primarily interacting with copied templates during exponential amplification (*Green, Venkatramanan & Naqib, 2015*; *Naqib, Poggi & Green, 2019*). By performing DePCR with synthetic DNA templates with known primer site sequences and recognition sequences, the sequences of primers that anneal to templates during linear copying are preserved and the number of mismatches between primer and template can be determined by sequencing (Fig. 2). Each DePCR sequence read generated in this study has two pieces of information: (a) the template that was copied (identified from the recognition sequence), and (b) the primer annealing to the template during the initial two cycles of linear copying. For standard PCR, each sequence read generated in the study also provides information about which template was copied but does not provide information about which primer annealed to the template during the initial two cycles of linear copying. Instead, standard PCR shows the sequence of a primer that annealed to a copy of the original template at some stage during the 28 cycles of amplification. As part of this study, we initially performed 16 different experiments to examine the effects of PCR amplification method (PCR or DePCR) and annealing temperature (45 °C or 55 °C) (Figs. S1–S16). Each experiment was a PCR amplification of synthetic DNA templates, ranging from a single template to a combination of up to 10 different templates. In some experiments, synthetic DNA templates ('ST's) were added to the PCR reaction mixture at equimolar

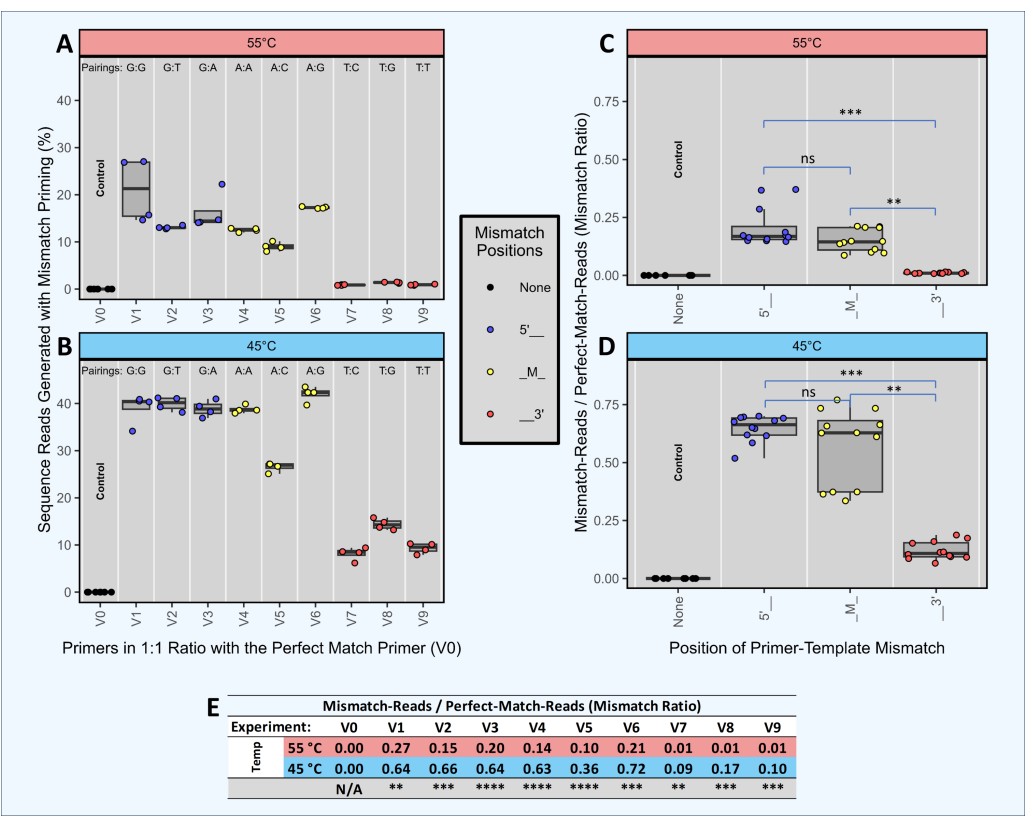

**Figure 3** **Head-to-head competition between perfect match primers and single-mismatch primers when amplifying a single template.** Synthetic template ST0 was amplified using DePCR in reactions with two primers only: a perfectly matching primer (V0) and a primer with a single mismatch. Perfect match and mismatch primers were input into the reaction at equimolar levels, and amplifications were performed using 45 °C or 55 °C annealing during the 1st stage of DePCR. For each primer combination, four technical replicates were performed. Using the scheme depicted in Fig. 2, the sequenced amplicons were then evaluated to determine which primer annealed to the ST0 template during the first two cycles of the DePCR reaction and to determine the level of perfect match and single mismatch annealing and elongation. Control reactions were conducted wherein only perfectly matching primers (V0) were used. The percentage of sequences with mismatch primers are shown in (A) (55 °C) and (B) (45 °C). Experiments conducted with primers containing mismatches towards the 5′ end of the primer are colored blue, while those with mismatches towards the 3′ end of the primer are colored red. Experiments with mismatches in the middle position ('M') are colored yellow. The mismatch pairing (template:primer) is shown above each column. The ratio of mismatch-to-perfect-match primer usage by mismatch location is shown in (C) (55 °C) and (D) (45 °C). An ANOVA was run for each annealing temperature followed by Tukey's post hoc test for pairwise comparison between mismatch positions. The exact values of the mismatch-to-match ratios are shown in the table (E), with significance values from Welch's two-sample-$t$-test comparing results from experiments with different annealing temperatures. Significance levels: 'ns' ($p \geq 0.05$), '*' ($p < 0.05$), '**' ($p < 0.01$), '***' ($p < 0.001$), '****' ($p < 0.0001$).

concentration, while in others, each template was added at a different concentration. In addition to varying input templates, 64 primers were used in different combinations to amplify the STs. In some reactions, only a single primer was used, while in most reactions, combinations of 64 primers were used. When multiple primers were used, they were present in equimolar concentration. A summary of experimental conditions is shown in Table S6.

7–8 technical replicates were generated for each experimental condition. We repeated some experiments with proofreading polymerases to determine if polymerase errors contributed substantially to the observed patterns of primer utilization and template abundance. Finally, we performed a series of experiments in which a single template was amplified with a mixture of perfect match and single mismatch primers present at equimolar ratio (experiment A+) to evaluate the impact of mismatch position and nucleotide pairing on amplification efficiency.

### Impact of mismatch position on single template amplification in head-to-head competition between two primers (Experiment A+; Fig. 3)

We performed an experiment with multiple independent reactions in which a single gBLOCK template (template ST0) was PCR amplified with two different primers in equimolar concentration. Each reaction contained one primer that matched the template perfectly (V0) and another primer at equimolar concentration that had a single mismatch with the ST0 template (*i.e.,* primers V1 through V9). Control reactions with the V0 primer alone were also performed. These experiments were performed only using DePCR and with the QB mastermix. Each of the primers used contained a single mismatch relative to the template, but the position of the mismatch varied (5′, mid (M), or 3′) as did the base-pairing at each position. For example, primers V1, V2 and V3 each have a different 5′ position mismatch with the ST0 template (G, T, and A, respectively). The purpose of this experiment was to determine the effects of primer mismatch position and sequence on primer annealing to templates. Reactions were performed at 45 °C and 55 °C to further examine the role of annealing temperature, with the assumption that reactions conducted at lower annealing temperature would be more tolerant of mismatches.

The results indicate that position of mismatch, annealing temperature, and to a lesser extent mismatch nucleotide sequence, have a significant effect on template amplification. Under conditions where mismatches do not affect primer annealing, amplification of the single template by the perfect match and mismatch primers should be roughly equivalent (*i.e.,* ratio of mismatch-to-match amplifications = 1). Using DePCR, the ratios of mismatch-to-perfect match amplifications ('mismatch ratio') were found to vary from 0.01 to 0.72, depending on annealing temperature, position of mismatch and base mismatch (Fig. 3). At low annealing temperatures and 5′ mismatches (V1–V3), mismatch pairing was well tolerated, regardless of sequence (mismatch ratio of 0.64–0.66); this was significantly reduced at the higher annealing temperature of 55 °C (mismatch ratio of 0.15–0.27). Mid-position mismatches (V4–V6) were similarly affected by annealing temperature (mismatch ratio of 0.10–0.21 at 55 °C and 0.36–0.72 at 45 °C) and to a certain extent by nucleotide mismatch (low mismatch ratio for primer V5 (C-A pairing) relative to primers V4 (A-A pairing) and V6 (G-A pairing)). 3′ mismatches (V7–V9) were particularly inhibitory for amplification, regardless of nucleotide base, but were strongly affected by annealing temperature (mismatch ratio of 0.09–0.17 at 45 °C and 0.01 at 55 °C).

Thus, using DePCR and our synthetic DNAs in a simple experimental system with only two primers at a time, we demonstrate empirically the negative effects on annealing of mismatches between primer and template, and that these negative effects can be partially

ameliorated using lower annealing temperature. Similar to prior studies (*Bru, Martin-Laurent & Philippot, 2008*), we observed that mismatches close to the 3′ end are heavily destabilizing, likely due to difficulty in binding of polymerase. Despite the destabilizing nature of 3′ mismatches, at a low annealing temperature of 45 °C, about 10% of template copies were made with 3′ mismatch primers, indicating that low annealing temperatures can help improve mismatch amplifications (*Ishii & Fukui, 2001*). Mismatches more distal from the 3′ end affected the overall primer annealing/elongation to a much smaller extent, particularly at lower annealing temperatures—suggesting that in complex reactions with mixed templates and degenerate primers, where mismatches are likely to be abundant, annealing temperatures should be optimized to be as low as viable. Consistent with the results we present here from a simple synthetic DNA system, we previously observed that in a complex fecal sample amplified with an 18-fold degenerate primer, the diversity of primer utilization in amplifying gDNA template was inversely correlated with annealing temperature (*Naqib, Poggi & Green, 2019*). Thus, we demonstrate that while the primer annealing-polymerase elongation system disfavors mismatches, the strength of the effect is dependent on mismatch position and annealing temperature and can also be influenced by mismatch sequence pairing. We further note that the magnitude of the effect is likely to vary between PCR primer sets, as well as by PCR conditions such as mastermix composition, polymerase and annealing temperature.

## DePCR outperforms standard PCR when characterizing mock community DNA using simple and complex primer pools (Fig. 4)

We previously demonstrated that the DePCR method can be used to characterize mock DNA mixtures, and that representation of the ratio of the underlying DNA templates is improved (*Green, Venkatramanan & Naqib, 2015*). In the prior study, only 4 templates were utilized, and a single degenerate primer set. To further explore primer-template interactions in more complex systems that better approximate the amplification of mixed templates using non-degenerate and degenerate primer sets, we developed a series of experiments in which primers and templates were mixed in different ratios and amplified using standard PCR and DePCR (Figs. S1–S16; Tables S2 and S6). In this manuscript, we focus on a series of experiments (*i.e.,* "B" experiments), in which 10 synthetic DNA gBlocks were mixed in equimolar ratios and used as templates for amplification with different primer pools (Fig. 4 and Figs. S6–S9). Tested primer pools included a single primer perfectly matching one of the 10 gBlocks and having a single mismatch with each of the other 9 gBlocks (experiment B1), ten primers each perfectly matching a single template and having one or two mismatches with the other nine gBlocks (experiment B10), and 27 primers with one to three mismatches with each of the 10 gBlocks (experiment B27). Amplifications were performed at 45 °C and 55 °C using standard PCR and DePCR, and results were visualized in an ordination plot (Fig. 4). The ratio of mismatch-to-match amplifications ('mismatch ratio') was calculated for each reaction.

The results of these experiments demonstrate that DePCR was superior to standard PCR for representing the underlying template mixture after PCR amplification and sequencing (Fig. 4). Visually, the MDS plot shows all DePCR samples closer to the input ratio of
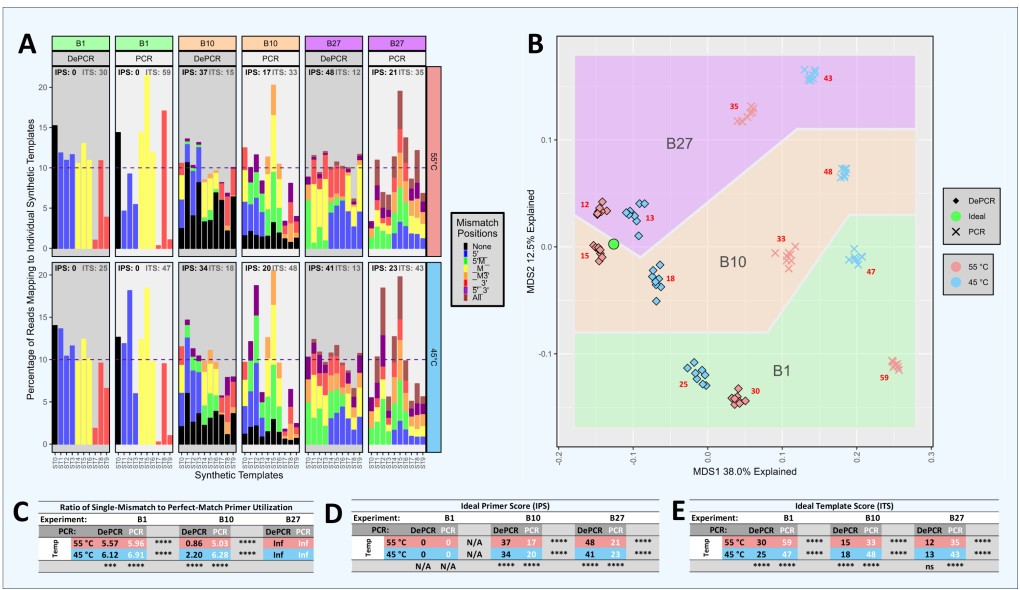

**Figure 4  Amplification and sequencing of template pools with primer pools of varying complexity.** Ten synthetic DNA templates (ST0 through ST9) were pooled equimolarly and amplified using standard PCR and DePCR with different primer combinations. These conditions included: a single primer perfectly matching the ST0 template (primer V0; 'B1' experiment), ten primers each perfectly matching one of the STs (primers V0–V9; 'B10' experiment), and 27 primers with one to three mismatches with each of the ten templates and no perfect matches (primers V10–V36; 'B27' experiment). Each condition was performed with eight replicates. (A) For each experiment (B1, B10, B27) and for each template (ST0 through ST9), the distribution of primers annealing to each template are shown for PCR and DePCR at 45 °C (bottom) and 55 °C (top) annealing temperature. The distributions are color-coded by the type of primer-template interaction, including perfect matches (black) and single, double or triple mismatches (color coded by locations of mismatches). The dotted line represents the 10% relative abundance of each ST added to the reaction. Metrics for evaluating how evenly the primers were utilized and templates amplified are shown at the top of each panel and in tables below. These include the Ideal Score for primers (IPS; black) and the Ideal Score for templates (ITS; grey). For both ITS and IPS, values range from 0 (perfect representation of input) to 200. IPS and ITS values are shown in tables D and E respectively, with significant differences by annealing temperature or by PCR method indicated. Ratios of mismatch amplifications to perfect match amplifications (mismatch-to-match ratio) are shown in table C. Mismatch-to-match calculations could not be performed for the B27 experiment due to the absence of perfect match primers (n/0 = Inf). Visualization of the recovered template profiles relative to the known input template composition was performed with a multi-dimensional scaling plot ordination (B). The green circle represents the input template composition, and the data points closer to the green circle indicate closer representation of the experimental results compared to the input templates. Data from the B1, B10 and B27 experiments are shown for standard PCR and DePCR at 45 °C and 55 °C annealing temperatures. The red numbers indicate the average ITS score for each condition. All $p$-values were calculated with Welch's-two-sample-$t$-test. Significance levels: 'ns' ($p \geq 0.05$), '*' ($p < 0.05$), '**' ($p < 0.01$), '***' ($p < 0.001$), '****' ($p < 0.0001$).

templates (*i.e.*, green circle), regardless of primers used or annealing temperature. At 55 °C annealing temperatures, ideal template scores (ITS) for standard PCR ranged from 33 (10 primers) to 59 (one primer), while those for DePCR ranged from 12 (27 primers) to 30 (one primer). At 45 °C annealing temperatures, ITS values for standard PCR ranged from 43 (27 primers) to 48 (10 primers), while those for DePCR ranged from 13 (27 primers) to 25 (one primer). In experiment B10 (10 primers × 10 templates), higher annealing

temperatures yielded slightly improved results (ITS values of 15 *vs.* 18 with DePCR and 33 *vs.* 48 with standard PCR). For DePCR, 27 primers without any perfect matches with the 10 templates, yielded the lowest ITS values (12–13) regardless of annealing temperature.

The use of primers with mismatches was affected by the primer pool and by the annealing temperature employed for standard PCR and DePCR. In experiment B1, with a single primer, mismatch amplifications dominated with mismatch-to-match ratios ranging from 5.57 to 6.12 across both annealing temperatures using DePCR (Fig. 4). Since the primer employed was a perfect match for only a single template, while the remaining nine templates had one mismatch each with the primer, a high ratio of mismatch amplification was expected. When 10 primers were employed, each matching one of the ten templates, the mismatch ratio was significantly lower for DePCR (0.86 at 55 °C and 2.2 at 45 °C). Values measured for standard PCR are not reliable for these measurements because they do not reflect the annealing of primers to original templates but rather primers annealing to copies of the original templates. In standard PCR, the copies do not contain the original template primer site sequence, but rather the sequence of whichever primer is annealed to the template. The high utilization of perfect match primers in the B10 experiment was expected, as a single perfect match primer exists for each template. Nonetheless, even when using 10 templates with 10 perfectly matching primers, mismatch annealing predominates at lower annealing temperatures (*e.g.*, mismatch ratio of 2.2 at 45 °C in experiment B10). Thus, despite annealing-and-polymerase extension heavily favoring perfect matches in simple primer-template reactions (as shown in Fig. 3), more complex template-primer interactions (Fig. 4, experiment B10) appear to enhance the role of mismatch annealing, even when adequate concentrations of perfect match primers are present. This effect is enhanced at lower annealing temperature. Somewhat surprisingly, in the presence of a more complex pool of primers but without any perfect match primers (experiment B27), the DePCR reaction was best able to approximate the underlying template pool (Fig. 4). Under these conditions, temperature was not a significant factor and single mismatch and double mismatch annealing and extension predominated, while triple mismatches were rare.

These results demonstrate that with a complex template pool: (a) DePCR consistently outperformed standard PCR, regardless of primer pool employed (Fig. 4 and Figs. S6–S9); (b) mismatch amplifications appear to be strongly favored when complex primer pools are employed, and that this phenomenon is enhanced at low annealing temperatures; (c) complex pools of primers can yield improved outcome relative to simple primer pools, even without perfect matches; and (d) higher annealing temperatures were only preferred when perfectly matching primers were available for all templates—a rare occurrence in 16S rRNA gene amplicon studies. Based on these data, we conclude that DePCR can be used to improve recovery of underlying template ratios in the presence of simple and complex primer pools, and that perfectly matching primers are not required for amplification of complex template pools.

## Role of mismatch amplification with simple and complex primer pools (Fig. 5)

We further explored the role of mismatches, mismatch position, and multiple mismatches on template amplification in an experimental system with only a single template (Fig. 5; Tables S2 and S6). We examined primer utilization patterns in reactions with a single template (ST0) amplified using a equimolar mixture of 10 different primers (experiment A10; 0–1 mismatches) and using an equimolar mixture of 64 different primers (experiment A64; 0–3 mismatches). Experiments were performed with standard PCR and with DePCR. Since there is only a single template, the data output from these experiments represent the relative abundance of each primer used to amplify the single template, ST0. In DePCR, these data represent the primers that annealed to the ST0 template during the first two cycles of PCR (*i.e.,* linear copying). In standard PCR, these data represent the final distribution of primer usage after 28 cycles of amplification, and do not represent the primers that annealed to the ST0 template during linear copying stages. We use the Ideal Primer Score (IPS) as a metric to measure evenness of primer use in standard PCR and DePCR, with low IPS values indicating greater evenness of the primer distribution in the output sequence data.

Thus, in standard PCR (low IPS scores), we observed that primers were used more evenly and showed less specificity for better matching templates. We term this phenomenon 'scrambling' due to loss of information as these data are derived from primers annealing to PCR copies rather than the original ST0 template. In DePCR, primer utilization is quite uneven because primer-template reactions are selective based on mismatch presence and location, and this information is retained during exponential amplification. This can be observed by looking at the IPS values which indicate a greater unevenness (*i.e.,* higher ideal scores) relative to standard PCR. In standard PCR, due to the scrambling of the signal of primer-DNA template interactions, primers with multiple mismatches were readily utilized during PCR cycling. For example, primers with mismatches at all three positions relative to the single DNA template represented ∼20% of all template copies when 10 primers were used, and >30% of all template copies when 64 primers were used. When 10 primers were used with standard PCR, perfect match copies represented approximately the percentage of primers added to the reaction that had a perfect match (*i.e.,* 10%). Little difference was seen between 45 °C and 55 °C annealing temperatures, for both A10 and A64, as indicated by IPS values.

Conversely, when using DePCR to amplify the ST0 template with 10 primers (A10 experiment), perfect match annealing (*i.e.,* primer V0) was observed in 20–25% of reads while the remaining reads were generated with single mismatch annealing, regardless of annealing temperature. Primers with single mismatches dominated due to the nine-fold higher abundance of mismatch primers (primers V1–V9) relative to perfect match primers (V0) in the reaction. The mismatch ratio (ratio of mismatch-annealing to perfect match annealing) ranged from 3.11 to 3.42, depending on annealing temperature. Primers with mismatches at the 5′ position were most favored, while 3′ mismatches were the least favored. Tolerance for 3′ mismatches was slightly improved with lower annealing temperature. The more complex primer pool (64 primers; experiment A64) was affected more strongly

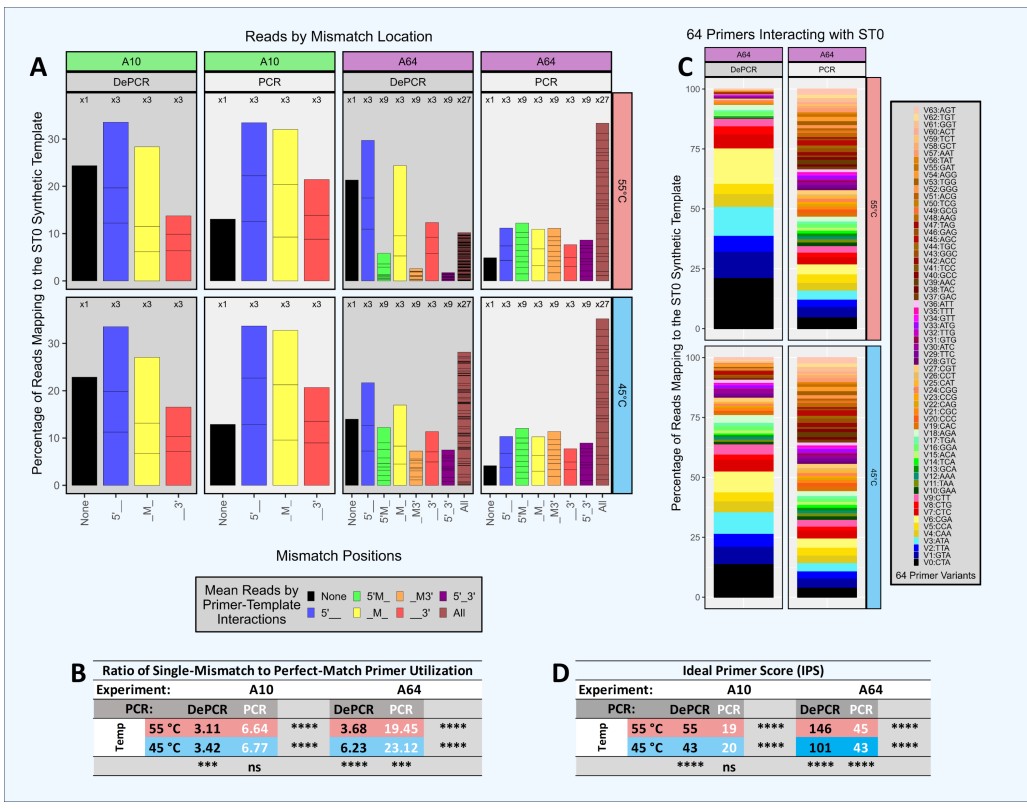

**Figure 5** **Amplification and sequencing of a single template with primer pools of varying complexity.** A single synthetic DNA template (ST0) was amplified using standard PCR and DePCR with either ten primers (primers V0–V9; 1 perfect match and nine single mismatch primers; 'A10' experiment) or 64 primers (primers V0–V63; 1 perfect match and 63 primers with one to three mismatches; 'A64' experiment). All experiments were conducted with eight replicates. (A) For each experiment, the distribution of perfect match and mismatch amplifications is shown with color coding of bar charts to indicate type of mismatch amplification. For example, perfect matches are shown in black, while triple mismatch amplifications (27 possible combinations) are shown in brown. The number of possible mismatch types are indicated above each column (one, three, nine, or 27 possible combinations). (B) The ratio of mismatch amplifications of any type to perfect match amplifications is shown in the table below. (C) For experiment A64, the distribution of the usage of all 64 primers is shown for standard PCR and DePCR at 45 °C (bottom) and 55 °C (top). (D) The primer distributions in the sequencing output were compared to the equimolar input of the 10 (experiment A10) or 64 primers (experiment A64) by calculating the Ideal Score for Primers (IPS) for each replicate and subsequently averaged. Large IPS values represent more selective utilization of primers (*i.e.*, more uneven), while small IPS values represent more broad utilization of all primers provided. Primer distributions from standard PCR do not represent primer-template interactions but represent the less informative primer-amplicon interactions. The primer distributions in DePCR represent the primers annealing to templates during the first two cycles of template copying. Welch's two-sample $t$-test was used to calculate significance values for annealing temperature effects and separately for PCR method effects. Significance levels: 'ns' ($p \geq 0.05$), '*' ($p < 0.05$), '**' ($p < 0.01$), '***' ($p < 0.001$), '****' ($p < 0.0001$).

by annealing temperature. The mismatch ratio in DePCR ranged from 3.68 (55 °C) to 6.23 (45 °C), and this was due to greater tolerance for double- and triple-mismatch primer-template annealing events at the lower annealing temperature.

By increasing the complexity of the primer pool, the relative abundance of primers with perfect matches decreased (experiment A64). This appears to make the overall system more sensitive to annealing temperature, and with lower annealing temperature, allowing primers with multiple mismatches or 3′ mismatches to become more involved in template annealing. Given the likelihood of mismatch annealing events dominating in amplification reactions of natural samples with high template sequence diversity, increased primer degeneracy may be helpful provided that the increasingly degenerate primers are still able to anneal to the region of interest and not to multiple other locations across the genome. In addition, as DePCR appears to increase tolerance for mismatches (*Green, Venkatramanan & Naqib, 2015*; *Naqib, Poggi & Green, 2019*), it may also be viable to reduce the number of degenerate primers, particularly those with wobble bases at 5′ and middle positions, when designing highly degenerate primers for functional gene analysis. This could lead to streamlined primer pools with broader coverage.

## Contribution of polymerase error to primer utilization profiles (Fig. 6)

Our initial studies (*e.g.*, Figs. S1–S16; Figs. 4 and 5) were performed with a standard, non-proof reading *Taq* polymerase (MT). We observed that a certain percentage of reads were identified as containing primer-template interactions that could not exist based on the primers and templates added to the reaction. For example, in experiment A1 (Fig. S1), only a single template (ST0) and a single primer (V0; no mismatches with ST0 template) were added. This control experiment should only have resulted in perfect match annealing, but roughly 4% of reads contained a mismatch. As primer and template pools become more complicated, our ability to identify incorrectly assigned reads becomes confounded. We hypothesized that the incorrectly assigned reads are a result of polymerase errors incorporated during amplification steps in DePCR. In standard PCR, the sequence of the primer site is not informative because the DNA at the primer site is the actual synthetic oligonucleotide primer. In DePCR, the region of the primer site is a copy of the oligonucleotide primer that annealed to the template during the initial two cycles of copying. Subsequently, exponential amplification is performed using the Illumina P5 and P7 primers, and the polymerase must copy through the original priming site, thus providing opportunities for the polymerase to make errors. Polymerase errors in the priming site region contribute to misidentification of the utilized primer.

To address the concern that polymerase errors contribute substantially to the primer utilization profile signature, we repeated some experiments with proof-reading polymerases and primers containing phosphorothioate linkages to prevent polymerase exonuclease activity on the primers themselves. Proof-reading polymerases were used to repeat experiment A1 as this simple experiment (single template and single primer) can most easily be used to identify polymerase errors. We compared error rates in the bases of the primer site itself (positions 1–20), with the next 20 bases (positions 21–40), and the next 60 bases after that (positions 41–100). Experiments were conducted using standard PCR and DePCR at 45 °C and 55 °C, and with a proof-reading polymerase (QB) and a non-proof-reading polymerase (DT). These results were compared to the prior analysis

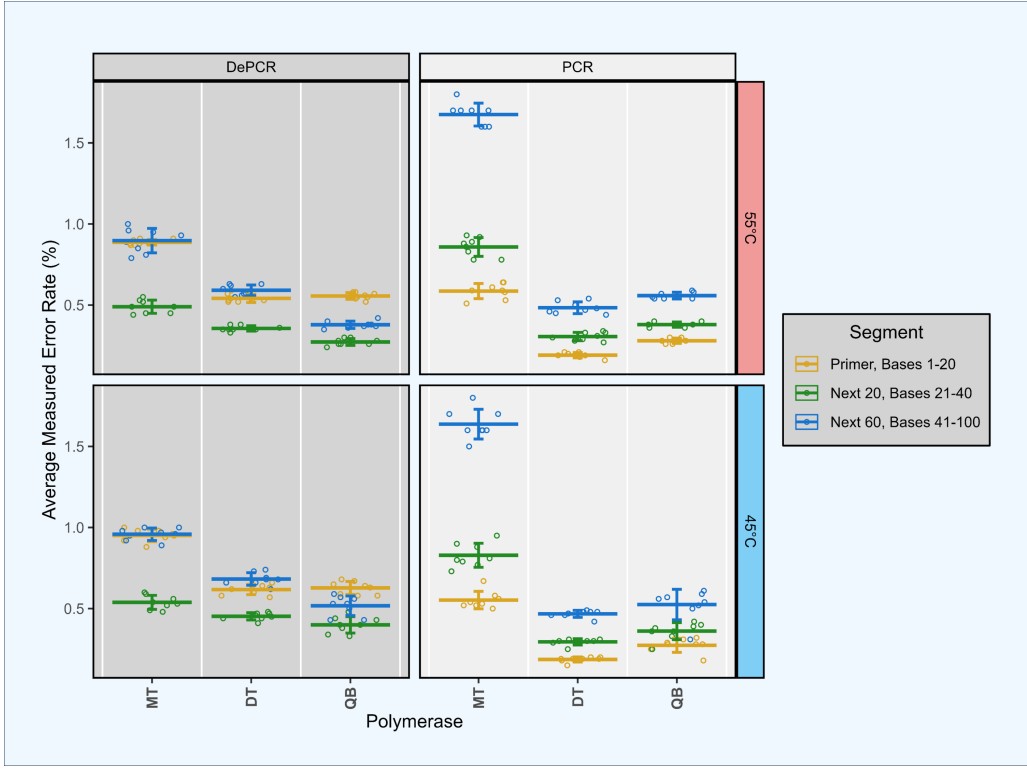

**Figure 6** **Estimation of polymerase error rates in standard PCR and DePCR using proofreading and non-proofreading polymerases.** Synthetic template ST0 was amplified using a perfectly matching primer (primer V0) in standard PCR and DePCR reactions with three different PCR mastermixes. The three mastermixes included MT (non-proofreading), DT (non-proofreading) and QB (proofreading). DT and QB reactions were conducted with primers containing phosphorothioate modifications to prevent exonuclease activity of proofreading polymerases. Reactions were conducted at 45 °C (bottom), and 55 °C (top) annealing temperatures with seven to eight replicates. Error rates were measured by mapping sequence data against the ST0 reference and the number of mismatches between reference and sequence averaged across the first 20 bases (location of the primer), the next 20 bases (positions 21–40), and the next 60 bases (positions 41–100). Mean and standard deviation are shown. The first 20 bases (primer site) had significantly lower error rates in standard PCR relative to the next 20 bases ($P < 2.93e{-}03$). Conversely, the first 20 bases had significantly higher error rates in DePCR relative to the next 20 bases ($p < 1.46e{-}07$).

conducted with a non-proof-reading polymerase (MT) and with standard primers without phosphorothioate modifications (Fig. 6).

This analysis demonstrated that the original analysis, performed with the MT polymerase, had substantially higher error rates (0.74–1.4%) than observed in the second analysis with both proof-reading and non-proof-reading polymerases with the phosphorothioate modified primers (0.27–0.66%; $p < 7.89e{-}08$). Using phosphorothioate modified primers, error rates were only slightly different between proof-reading (0.27–0.59%) and non-proof-reading polymerases (0.33–0.66%; $p < 4.78e{-}02$; Welch's two-sample-$t$-test) across the first 100 bases of sequences.

We examined error rate by location as well, as the first 20 bases in standard PCR represent the synthesized primer, while the first 20 bases in DePCR represent PCR copies

of the primer. In all polymerase and annealing temperature experiments, error rates in DePCR in the first 20 bases were significantly higher relative to standard PCR in the first 20 bases ($p < 4.99e{-}08$; Welch's two-sample-$t$-test). No consistent trends were observed for the next 20 bases after the primer region between PCR and DePCR, or for the next 60 bases after that.

Surprisingly, in DePCR experiments, the first 20 bases (*i.e.,* that of the primer site itself) always had a significantly higher error rate than the next 20 positions ($p < 1.46e{-}07$; Welch's two-sample-$t$-test). Positions 41–100 had error rates either in between the 1–20 and 21–40 positions, or roughly the same as the 1–20 positions. The cause of the higher error rate in bases 1–20 is not clear, but may have to do with lower quality on Illumina sequencers during the initial cycles of sequencing, leading to higher accuracy after the first 20 bases, and then a more gradual decline in quality as the run continues. Conversely, in standard PCR, this trend was inverted, with error rates in positions 1–20 significantly lower than for positions 21–40 ($p < 2.93e{-}03$; Welch's two-sample-$t$-test).

Thus, although the polymerase does introduce some error into the primer site during DePCR, and this does introduce error into primer utilization profiling, the magnitude of the error appears to be manageable (<5% misassignment, as evaluated in experiment A1 with a single template and a single perfectly matching primer; Fig. S1). The error rate does appear to be further reduced with phosphorothioate modified primers, though there was no substantial difference between proof-reading and non-proofreading polymerases when using such primers.

## CONCLUSIONS

PCR bias has been thoroughly studied, and a wide range of contributing factors are known. PCR selection—wherein factors within PCR preferentially amplify some templates (*Polz & Cavanaugh, 1998*)—can strongly distort underlying biological structure. We focus in this study on primer-template interactions, as mismatches are known to lead to PCR selection (*Mao et al., 2012*; *Reysenbach et al., 1992*). As has been shown previously, templates with mismatches to primers can be difficult to detect, and mismatches close to 3′ ends are particularly problematic for template amplification (*Bru, Martin-Laurent & Philippot, 2008*; *Wu, Hong & Liu, 2009*). Using Deconstructed PCR (DePCR), we previously demonstrated that the method can substantially reduce bias in PCR amplification of mock DNA mixtures and is more tolerant of mismatches between primer and template than standard PCR (*Green, Venkatramanan & Naqib, 2015*; *Naqib, Poggi & Green, 2019*). Furthermore, we found that while perfect match annealing and extension is strongly favored in simple head-to-head competition between mismatch and perfect match primers, perfect match annealing and extension are less favored in complex template and primer pools. Complex primer pools without any perfect matches can robustly amplify a complex mixture of templates using DePCR. For complex systems, lower annealing temperatures are favored though higher annealing temperature can succeed when perfect match primers and templates are present. Thus, we demonstrate that DePCR can serve as a new tool to interrogate how primers and primer pools interact with DNA templates in an empirical

manner. This tool has relevance to PCR optimization for amplifications where mismatches between primers and templates are likely, including microbial 16S ribosomal RNA gene and functional gene amplification, targeted amplification of bisulfite converted DNA, or in amplifications of rapidly mutating regions of viral genes.

## ACKNOWLEDGEMENTS

We gratefully acknowledge the support of the members of the Genome Research Core (GRC; University of Illinois at Chicago) and the Genomic Microbiome Core Facility (GMCF; Rush University) for assistance with this study.

### Funding

The authors received no funding for this work.

### Competing Interests

The authors declare there are no competing interests.

### Author Contributions

- Jeremy Kahsen performed the experiments, analyzed the data, prepared figures and/or tables, authored or reviewed drafts of the article, and approved the final draft.
- Sonia K. Sherwani performed the experiments, analyzed the data, prepared figures and/or tables, authored or reviewed drafts of the article, and approved the final draft.
- Ankur Naqib conceived and designed the experiments, performed the experiments, analyzed the data, prepared figures and/or tables, authored or reviewed drafts of the article, and approved the final draft.
- Trisha Jeon performed the experiments, authored or reviewed drafts of the article, and approved the final draft.
- Lok Yiu Ashley Wu performed the experiments, authored or reviewed drafts of the article, and approved the final draft.
- Stefan J. Green conceived and designed the experiments, analyzed the data, prepared figures and/or tables, authored or reviewed drafts of the article, and approved the final draft.

### Data Availability

The raw sequence data files are available at the Sequence Read Archive (SRA) of the National Center for Biotechnology Information (NCBI): PRJNA513137. The data from A1 experiments conducted with DreamTaq and QuantaBio mastermixes and from experiment A+ are available at the SRA: PRJNA1072695.

The scripts used for analyses are available in the Supplemental Materials.

## Supplemental Information

Supplemental information for this article can be found online at http://dx.doi.org/10.7717/peerj.17787#supplemental-information.

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
