# Peer review of "Quantitating primer-template interactions using deconstructed PCR"

_PeerJ, doi:10.7717/peerj.17787_

## Round 0.1 · original submission · Major Revisions

I agree with the majority of the reviewers comments, particularly that you might want to clarify some sentences which might be obvious to you but for someone who does not have a deep knowledge the experimental designs it is difficult to follow. I am also the author of Suzuki and Giovannoni 1996 that you cite and I don't quite understand the relationship between our results and stochasticity (so perhaps clarify the statement). Our interpretation of the tendency of 2 template reactions to arrive at a 50-50% ratio was mostly that different templates reach the "plateau" at different points and rarer templates can "catch up" with more common templates.

Reviewer 1 ·

Basic reporting

Mostly acceptable, requires further clarification in a number of places. See additional comments below.

Experimental design

Needs to be reviewed by researchers with greater experimental knowledge than I have. From my perspective I see no overt issues.

Validity of the findings

Statsitics seem appropriate, some clarifications required. See additional comments.

Additional comments

In this manuscript, Kahsen et al. use mock libraries to provide an in-depth study of primer-template factors that drive PCR Bias. As a biostatistician with expertise in modeling PCR bias and limited experimental training, I find this work fascinating and impactful, yet also dense and difficult to review in some respects. From a biostatistical perspective, the analyses seem adequate and potential improvements to analytical methods are unlikely to modify the authors' core conclusions. Moreover, the data and insights the authors introduce will likely provide critical insights to future efforts to model/account for PCR bias during sequencing data analysis. While I have a few minor comments (e.g., data availability concerns) that should be easily addressed, my primary concerns relate to clarity of writing and interpretation of results. Note: I cannot knowledgeably evaluate the authors' experimental approach; another reviewer with experimental expertise is required for those aspects of this work.

What statistical tests (and multiplicity corrections) were applied? This comment applies to all figures with p-values.

Figure 3 is confusing; what does the "ratio of mismatch to perfect match priming" mean? What does it mean in the context of the V0 primer (as I understand it, a matched primer) to have zero reads in the left-hand panels? Is this just some weirdness with the definition of the vertical axis? Are they then taking ratios with those zeros to create the right panel? These terms/measures need to be better defined. Are these the same as the "mismatch ratio" discussed in the main text?

The section starting on line 407 and its accompanying figure (figure 5) were difficult to follow. I cannot follow the figure, and the main text makes it sound like the authors could identify (in standard PCR) which primers were binding to the template. I thought they said this was only a feature of DePCR. Also, how does "evenness" (assuming defined in the ecological sense) relate to their statements about scrambled primer usage patterns? This section and its accompanying figures need substantial work to be interpretable for a broad audience.

The authors' findings suggest that more complex pools of primers without perfect matches mitigate bias compared to perfect matches. This concept is discussed multiple times, but I find myself intrigued and curious. Is this a surprising phenomenon, or do the authors have some intuitive insights that could shed light on this? I believe further exploration of this aspect could add depth to the manuscript.

On Lines 480-483, the authors make an interesting observation but provide no context. Is this an expected result? Is there any intuition for why this might be occurring?

Minor:

"_M_" is not defined in the legend for Figure 3 (though it can be deduced from context). It's better to just mention it in the caption, though.

The authors' decision to make the rarefied tables available is a step in the right direction, but I would strongly encourage them to also provide the raw data. This could significantly enhance the impact of their work, as there are often better statistical approaches than rarefaction. The availability of raw data could open up new avenues for analysis and interpretation, potentially leading to even more profound insights. However, I must reiterate that the authors' analysis is likely sound in the context of the questions they were asking of this data, and I doubt alternative analytical approaches would greatly alter their core conclusions.

Reviewer 2 ·

Basic reporting

This is a very complicated and technical study. Overall, the manuscript is understandable and significant relevant data and references have been provided.

Experimental design

Overall, the experiments are well-designed. I have no comments on this.

Validity of the findings

All claims are supported by data. I have no comments on this.

Additional comments

1. The study is purely technical, and all the experiments were performed in well-designed 'artificial' conditions. In such conditions, all claims are well-supported by the data. However, I would wonder how biologically meaningful the conditions are? For example, in lines 361-372, the authors designed B1-B27 to prove that 'DePCR was superior to standard PCR for representing the underlying template mixture'. I wonder how well these designs align with real conditions researchers are going to face? It would be more biologically meaningful to point out the specific scientific question that requires using PCR to represent the template mixture and test the performance of DePCR and PCR in real conditions.
2. As I have mentioned above, I would be concerned about the application of this study as people may not bother reading a purely technical paper unless it benefits their own research. In lines 506-510, the authors talked about the biological relevance of the study in a very brief way. I would suggest expanding this part into a few paragraphs or maybe a whole discussion section to align the discoveries with biological questions.
3. In lines 302-316, the primer concentration used for mismatch/match is 200nM (according to the method section). I wonder whether the same trends will hold true if you increase or decrease the primer concentration while still preserving an equimolar concentration of match and mismatch primer.
4. To make the figures more organized, I would suggest splitting each figure into several panels like figure 1A, 1B... if possible and refer to each panel instead of the figure when describing results.

Reviewer 3 ·

Basic reporting

The paper has clear professional English language used throughout the manuscript.
Introduction shows enough context, but there are some statements where additional references will help (please see annotated PDF). The structure of the paper conforms to PeerJ standards. Figures are generally well labelled; however, some improvements are suggested in the annotated PDF.
Raw data and code availability section is missing in the main manuscript and need to be provided before acceptance. Scripts for analysis are currently part of supplemental, please include a link to access code and raw data in the main manuscript.

Experimental design

The design is well conducted and is within the scope of the journal.

This research describes the potential use of Deconstructed PCR as opposed to standard PCR in reducing PCR bias and demonstrates how DePCR performs better than standard PCR when mismatches between template and primer are present. This serves as a new tool to interrogate primer-template interactions in an empirical manner and has relevance to optimizing PCR amplifications where mismatches are likely. This is a useful technique of interest to the broad readership of PeerJ in terms of methodology in PCR applications.
However, the authors use a Rhodanobacter sp. as the only template source in their experimental design, and it might be worth including a discussion as to the choice of this template and if the results from using this species would ideally extend to other bacterial species.

The research questions are well defined and fills an identified knowledge gap.

Experiments are well described and investigation is well conducted as per the scope of the paper and is technically and ethically sound. Appropriate replications have been included in the analysis.

Methods are defined in sufficient detail and information to enable replication.

Validity of the findings

Underlying data needs to be provided, and some statistical analyses are suggested for clarity, please refer to annotated PDF.

Conclusions are well stated and linked to original research questions and limited to supporting results.

Annotated reviews are not available for download in order to protect the identity of reviewers who chose to remain anonymous.

---

## Round 0.2 · accepted · Accept

Hi one last comment that can be done at proofreading. There is a reference missing on line 277

Reviewer 1 ·

Basic reporting

appropriate

Experimental design

appropriate

Validity of the findings

appropriate

Additional comments

the authors have addressed my prior comments appropriately

Reviewer 2 ·

Basic reporting

All my concerns from the initial review have been addressed. I have no further concerns

Experimental design

All my concerns from the initial review have been addressed. I have no further concerns

Validity of the findings

All my concerns from the initial review have been addressed. I have no further concerns

Reviewer 3 ·

Basic reporting

N/A

Experimental design

N/A

Validity of the findings

N/A

Additional comments

The authors have improved the manuscript based on previous comments.